# Study on the Relationship between Richness and Morphological Diversity of Higher Taxa in the Darkling Beetles (Coleoptera: Tenebrionidae)

Liangxue Cheng [1], Yijie Tong [2], Yuchen Zhao [1], Zhibin Sun [3], Xinpu Wang [1,*], Fangzhou Ma [4,*] and Ming Bai [2,5,*]

1   School of Agriculture, Ningxia University, Yinchuan 750021, China; chengliangxue12@163.com (L.C.); 12021140135@stu.nxu.edu.cn (Y.Z.)
2   Key Laboratory of Zoological Systematics and Evolution, Institute of Zoology, Chinese Academy of Sciences, Box 92, Beichen West Road, Chaoyang District, Beijing 100101, China; tongyijie@ioz.ac.cn
3   National Space Science Center, Chinese Academy of Sciences, NO.1 Nanertiao, Zhongguancun, Haidian District, Beijing 100190, China; zbsun@nssc.ac.cn
4   National Key Laboratory of Environmental Protection and Biosafety, Nanjing Institute of Environmental Sciences, Ministry of Ecology and Environment, #8 Jiang-Wang-Miao-Jie, Xuanwu, Nanjing 210042, China
5   Hainan Yazhou Bay Seed Laboratory, Building 1, No.7 Yiju Road, Yazhou District, Sanya 572025, China
*   Correspondence: wangxinpu@nxu.edu.cn (X.W.); mfz@nies.org (F.M.); baim@ioz.ac.cn (M.B.)

**Abstract:** Many studies have found that the correlation between species richness (SR) and morphological diversity (MD) is positive, but the correlation degree of these parameters is not always consistent due to differences in categories and various ecological factors in the living environment. Based on this, related studies have revealed the good performance of using higher taxa in biodiversity research, not only by shifting the testing group scale from local communities to worldwide datasets but also by adding different taxonomic levels, such as the genus level. However, it remains unclear whether this positive correlation can also be applied to other categories or groups. Here, we evaluated the applicability of higher taxa in the biodiversity study of darkling beetles by using 3407 species (9 subfamilies, 89 tribes, and 678 genera), based on the correlation between taxa richness and morphological diversity in the tribe/genus/species. In addition, the continuous features prevalent in the tenebrionids, pronotum and elytron, were selected, and the morphological diversity of various groups was obtained by the geometric morphometric approach to quantify the morphologic information of features. This study found that genus/species richness in subfamilies Pimelinae and Stenochiinae was positively correlated with the change trend of MD, and the correlation between the MD of elytron and taxa richness gradually decreased from the tribe-level to the genus-level to the species-level test. The results confirm the stable morphology and simple function of the elytron and the applicability of tribe level in biodiversity research.

**Keywords:** biodiversity; higher taxa; darkling beetles; pronotum; elytron

## 1. Introduction

Species richness (SR), an important indicator to evaluate biodiversity, reflects the number of species in a community, geographic area, or evolutionary unit [1,2]; morphological diversity (MD) is also used as an indicator in biodiversity studies, reflecting biological evolution and ecosystem functions [1,3]. Many studies targeted at specific geographic populations have found a positive correlation between SR and MD [2,4–7]. Neustupa investigated the species diversity and morphological disparity of benthic Desmidiales in Central European peatland pools and found that they were highly correlated [7]; Bell investigated the species diversity, richness, and morphological diversity of sponges that experienced different water flow conditions at Lough Hyne Marine Nature Reserve (Ireland), found that MD was significantly positively correlated with both species diversity and richness, and

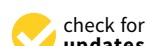



confirmed that it could be used as a qualitative predictor of species diversity and richness of sponges [4]. However, geographic populations are often affected by factors such as the environment and ecological niche of sample sites, leading to unstable relationships between biodiversity parameters, and some studies have found that the positive correlation between SR and MD is not applicable in all cases. By comparing the patterns of richness and morphodiversity of the genus *Porites* in the Pacific, Mohedano-Navarrete found that the relationship between SR and MD showed a logarithmic curve that tended to be asymptotic and nonlinear because of the saturation of niche availability and species boundaries [3]. Zhang evaluated the MD of the pronotum and elytron of 1303 stag beetles and found that the correlation between SR and MD was unstable in different test groups [8]. In addition, Foote found no correlation between the SR and MD of trilobites when comparing the morphological and taxonomic diversity of Blastoidea and Trilobita, as well as within the trilobite branch Libristoma, Asaphina, Proetida, Phacopida, and Scutelluina [9]. In recent years, studies on species biodiversity have gradually expanded from regional geographic populations to continent-level datasets in order to obtain more objective information by adding more diverse ecological factors and large-scale test indicators. Mindel analyzed the indicators of functional richness and divergence of benthic fish in the northeast Atlantic based on external morphological traits and found that the diversity indices were not always highly correlated by comparing the above indicators with the size diversity based on the individual body size and richness of the species [10]. Tong evaluated the relationship between genus/species-level richness and MD by using a worldwide dataset of jewel beetles and found that MD was positively correlated with genus richness but not significantly correlated with species-level richness, revealing the superiority of higher taxa in biodiversity [2]. However, the applicability of other higher taxonomic levels in biodiversity research has not been studied, and it remains unclear whether this positive correlation is always apparent in other groups.

　　Tenebrionidae, also known as darkling beetles, has about 2300 genera and 20,000 species all over the world, mainly in tropical, subtropical, and temperate regions and especially in equatorial regions [11–13]. The wide distribution, diverse habitats, and complex life history of darkling beetles allow this group to adapt to almost all biogeographical regions [12,14–18]. Their complex habitat in the process of evolution has also created the diverse external form of this population and has made this group an ideal research object of ecology and biogeography [19,20]. The pronotum, a common morphological feature of Coleoptera, connects the head with the mesothorax, which not only provides protection for the pterothorax but also plays an important role in the movement of the prothorax muscular system and the prothoracic legs. The elytron is a hard structure formed by highly specialized forewings that can protect the membranous hindwings and abdomen in addition to its flight function [21–24]. The rich ecological position and complex behavior of the tenebrionid make these two features become more important in their morphological evolution. Most studies focus on the taxonomy, molecular phylogeny, and biogeography of specific groups or geographic populations in Tenebrionidae [15,16,20,25–33]: Based on the phylogenetic analysis of abdominal terminalia and buccal apparatus, Aloquio revealed that the subfamily Diaperinae was polyphyletic, and the Nilioninae, which was once a tribe of Diaperinae, had been shown to be associated closely to Diaperini [34]; Kegoat analyzed the evolutionary history of 8 gene segments in 404 taxa of Tenebrionidae by using a variety of phylogenetic analysis methods, which strongly supported the monophyly of Tenebrionidae and highlighted that some important subfamilies and tribes may be paraphyletic or polygeneic, such as the subfamilies Diaperinae and Tenebrioninae. In addition, all of his phylogenetic analyses tend to support the traditional view that the family Tenebrionidae consists of three main lineages: the branches of 'Lagrioid', 'Pimelioid' and 'Tenebrionoid' [35–37]. While few studies pay attention to their biodiversity [2,8], the expression of higher taxa in biodiversity consisting of the phylogenetic results of this group is not yet clear. In this study, morphological information of test features was quantified by geometric morphometrics, and the MD index was obtained for comparison between taxa for

more objective comparability of test features. By using the number of tribes/genera/species in each living subfamily as the richness standard based on the worldwide Tenebrionidae dataset, the continuous features (pronotum and elytron) common to each tenebrionid group were selected to analyze the relationship between MD and SR and show whether the test parameters have consistent change rules and positive correlations among different higher taxa.

## 2. Materials and Methods

### 2.1. Samples Collection and Arrangement

In this study, standard dorsal view images of each subfamily of Tenebrionidae were used as test samples to construct a dataset, including 3407 species of darkling beetles covering 9 subfamilies (100% of the described subfamilies around the world), 89 tribes (93% of the described tribes), and 678 genera (Table 1). The continuous features, pronotum and elytron, were selected to evaluate the morphological diversity [38,39], and their morphological information in standard dorsal view was extracted based on geometric morphometrics. In addition, Nilioninae and Zolodininae, which have no tribe category, were tested as independent groups with other tribes in the tribe-level analysis [40].

**Table 1.** Sampling information of test subfamilies.

| Test Groups | Sampling Number of Tribes | Described Number of Tribes | Sampling Number of Genera | Described Number of Genera | Sampling Number of Species |
|---|---|---|---|---|---|
| Zolodininae | * | * | 2 | 3 | 2 |
| Lagriinae | 9 | 9 | 62 | 270 | 357 |
| Nilioninae | * | * | 1 | 1 | 21 |
| Phrenapatinae | 3 | 3 | 8 | 20 | 16 |
| Pimeliinae | 33 | 39 | 215 | 401 | 835 |
| Tenebrioninae | 29 | 29 | 220 | 955 | 1198 |
| Alleculinae | 2 | 2 | 54 | 140 | 317 |
| Diaperinae | 10 | 11 | 43 | 120 | 286 |
| Stenochiinae | 3 | 3 | 73 | 390 | 375 |
| Total | 89 | 96 | 678 | 2300 | 3407 |

Note: the '*' represents the subfamily without tribe.

The sample images used in the analysis were collected from the beetle collection at the National Zoological Museum of the Chinese Academy of Sciences and published monographs and literature [11,17,41–130], as well as taxonomic databases [131–148].

### 2.2. Richness Values and Morphological Diversity

In this study, a certain proportion of tribes (93% of the world's described tribes), genera (29% of the world's described genera), and species (17% of the world's described species) in each subfamily of Tenebrionidae were sampled for more objective analysis, and the number of samples in each test taxon was used as the richness value of the group (Tables 1 and 2) [11].

Geometric morphometrics were used to quantify the morphological information of the pronotum and elytron, according to the selection principles of homology, adequate representativeness, consistency of relative position, coplanarity, and repeatability [149]. The default shape of pronotum and elytron was symmetrical, and one contour curve was extracted from the left external contour of each (Figure 1A). The outer contour curve (curve 1) of the pronotum starts at the midpoint of the anterior margin and ends at the midpoint of the posterior margin; the outer contour curve of the elytron (curve 2) starts from the junction of the end of the scutellum and the leading edge of elytron and stops at the end of elytron (Figure 1B). Based on Tps-Dig (version 2.05) software, the shape information was digitized, and the outer contour curves of the test features were equally divided into 25 and 50 semi-landmarks with the same spacing to represent the morphological information of the

test site [150]. We used Tps-Small (version 1.2) [151] to test whether the observed variation in shape was sufficiently small that the distribution of points in the tangent space can be used as a good approximation of their distribution in shape space. Based on Morpho J 1.06a software, Procrustes variance was used to remove the non-shape differences caused by size, direction, and position by scaling, overlaying, and rotating the feature semi-landmark set of each test sample [152]. The variation trend of topological patterns formed by test features in geometric space was used to show the variation of morphological information (Figure 2) [2,153] through principal component analysis (PCA) in Morpho J (version1.06a) (Table 2, Figure 3) [154].

**Table 2.** Morphological diversity (MD) of test features in the subfamily test.

| Test Groups | MD of Pronotum | MD of Elytron |
|---|---|---|
| Zolodininae | 0.0046 | 0.0036 |
| Lagriinae | 0.0100 | 0.0039 |
| Nilioninae | 0.0130 | 0.0014 |
| Phrenapatinae | 0.0074 | 0.0026 |
| Pimeliinae | 0.0267 | 0.0051 |
| Tenebrioninae | 0.0124 | 0.0041 |
| Alleculinae | 0.0111 | 0.0019 |
| Diaperinae | 0.0295 | 0.0066 |
| Stenochiinae | 0.0171 | 0.0042 |

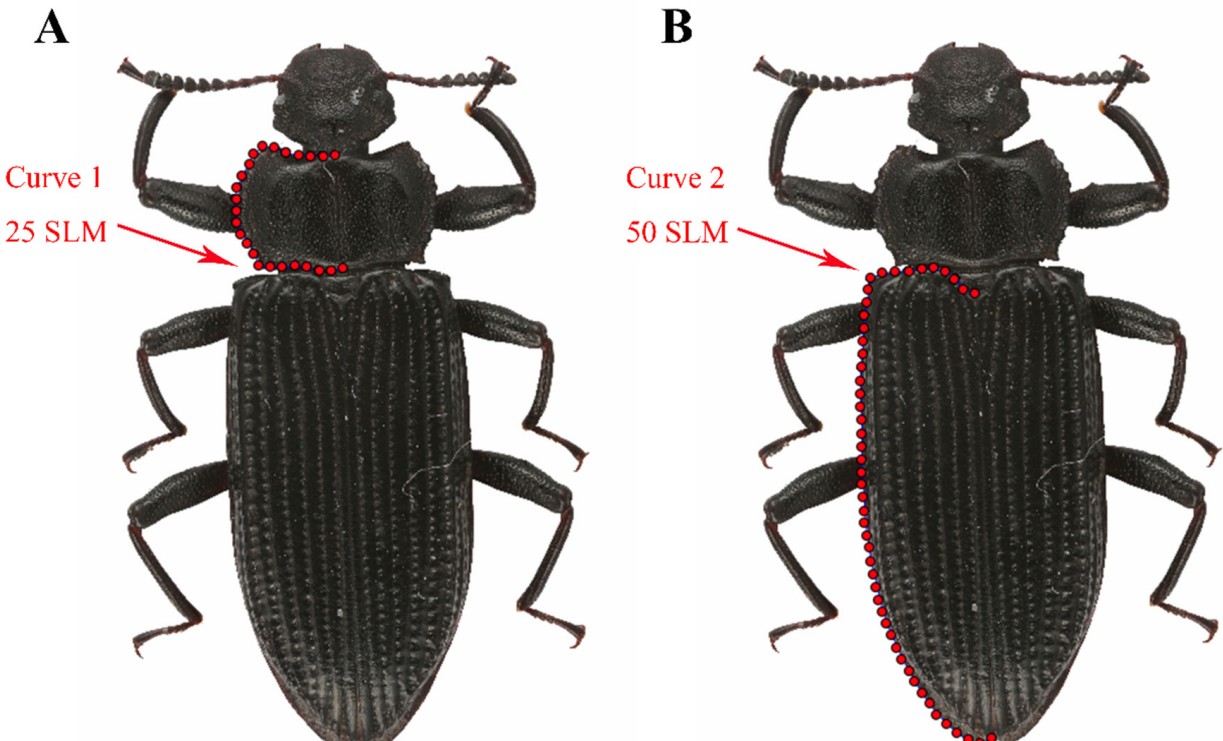

**Figure 1.** Semi-landmarks selection of test features based on Geometric Morphometrics, *Catapiestus* sp. (Tenebrionidae, Stenochiinae, Cnodalonini) was selected as an example: (**A**) Pronotum (curve 1) was resampled into 25 semi-landmarks; (**B**) elytron (curve 2) was resampled into 50 semi-landmarks.

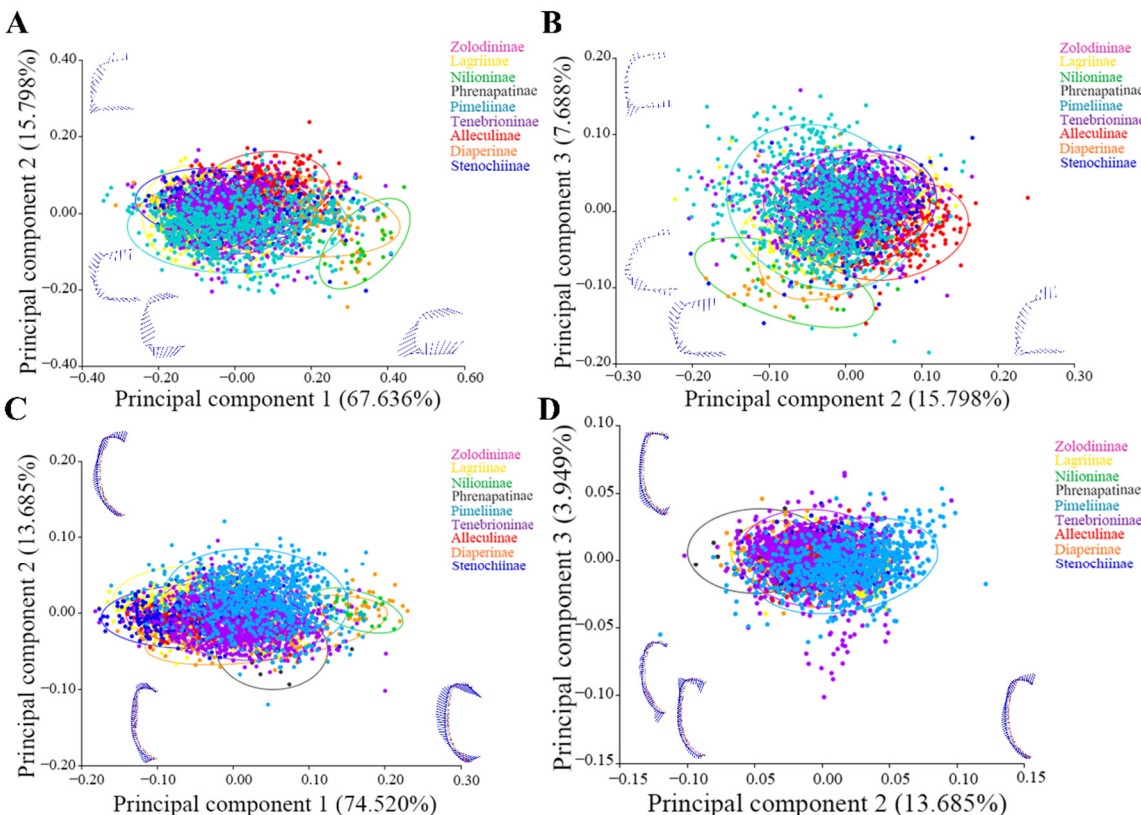

**Figure 2.** Principal component analysis on subfamily level: first three principal components of pronotum (91.122%) and elytron (92.154%) were selected to represent shape variation. (**A**) Variation of test pronotum in space built by PC1 and PC2. (**B**) Variation of test pronotum in space built by PC2 and PC3. (**C**) Variation of test elytron in space built by PC1 and PC2. (**D**) Variation of test elytron in space built by PC2 and PC3. (Note: Circles in figures were set to include at least 90% of samples in each test subfamily).

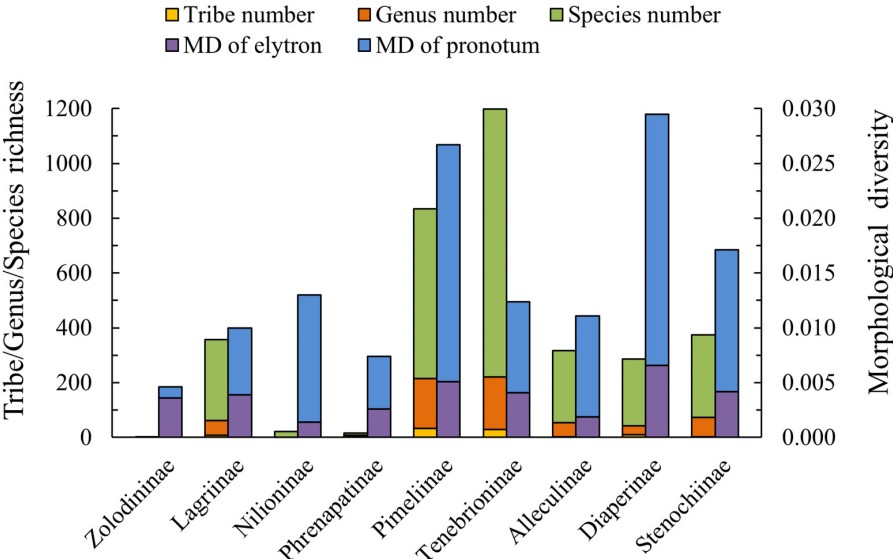

**Figure 3.** Morphological diversity and tribe/genus/species richness among subfamilies. Purple and blue columns represent morphological diversity of test pronotum and elytron. MD, morphological diversity. Yellow, orange, and green columns represent tribe, genus, and species richness, respectively, of each subfamily.

Using SPSS Statistics (version 26) [155], Spearman correlation coefficient (single-tail) analysis was performed to analyze the correlation between morphological diversity and taxa richness at the level of tribe/genus/species (Table 3, Figure 4A). In addition, the changes in correlation between the test parameters in the pronotum and elytron were compared by regression analysis, and the applicability of morphological information of different features to biodiversity research was evaluated (Figure 4B).

**Table 3.** Spearman correlation coefficient between morphological diversity (MD) and richness in different test taxa.

|  | Tribes Richness | | Genera Richness | | Species Richness | |
|---|---|---|---|---|---|---|
|  | r | p | r | p | r | p |
| MD of pronotum | 0.521 | 0.075 | 0.350 | 0.178 | 0.500 | 0.085 |
| MD of elytron | 0.782 | 0.006 | 0.600 | 0.044 | 0.571 | 0.077 |

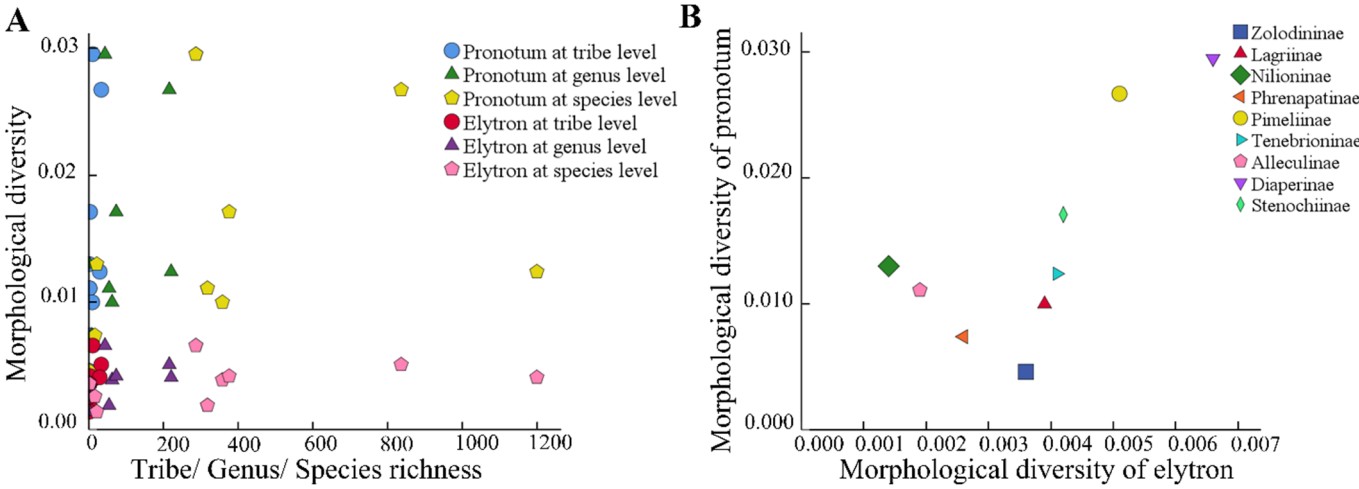

**Figure 4.** Correlation analysis of subfamilies of Tenebrionidae: (**A**) Relationships between morphological diversity and tribe/genus/species richness; (**B**) correlation between morphological diversity of pronotum and elytron.

## 3. Results

### 3.1. Morphological Variation of the Pronotum and Elytron

According to principal component analysis, the first three principal components (PCs) of the pronotum and elytron accounted for 91.122 and 92.154% of the morphological variation, respectively (Figure 2). For the deformation of the pronotum, along the positive direction of the first PC axis, the anterior and posterior margins obviously contract horizontally inward, the lateral margin expands outward, the base is nearly straight, and the overall contour is nearly rectangular; along the positive direction of the second PC axis, the anterior margin spreads obliquely outward, the lateral margin contracts inward, the posterior angle is obviously prominent, and the overall outline is nearly trapezoidal; along the positive direction of the third PC axis, the anterior margin shrinks inward, while the anterior and posterior angles expand outward, the deformation trend of the posterior angle is particularly obvious, and the lateral margin shrinks inward longitudinally and nearly straight, while the posterior margin shrinks inwardly, making the overall contour approximately square.

For the deformation of the elytron, along the positive direction of the first PC axis, the anterior margin shrinks inward, the anterior angle protrudes, the lateral margin expands in an arc from the middle forward, the elytron base shrinks inward, the elytron ends slightly protrude, the overall outline of the first half is broad, and from the middle to the end it narrows; along the positive direction of the second PC axis, the anterior margin expands

outward, the anterior angle shrinks inward, the shape variation is not obvious, the middle of the lateral margin expands outward and uplifts, the elytron base shrinks inward, and the overall contours become slightly wider; along the positive direction of the third PC axis, the anterior margin and front of the lateral margin of the elytron shrink inward though not obviously, the lateral margin back end expands outward, the elytron base significantly shrinks inward, and the overall profile is nearly parallel on both sides.

### 3.2. Taxa Richness and Morphological Diversity of Test Subfamilies

Analyses of the dataset using Tps-Small were performed to assess whether the amount of test shape variation in our dataset is small enough to justify the use of standard multivariate statistical methods in the tangent space. The correlation (uncentred) between the tangent space, Y, regressed onto the Procrustes distance (geodesic distances in radians) was 0.999935/0.999998 for the pronotum test and elyron group, respectively. There is little doubt on the basis of the results from Tps-Small, which supported the hypothesis that the prontum and elytron based on subfamily-level dataset can be analyzed by geometric morphometric methods since the results from the statistical test performed by Tps-Small proved the acceptability of the dataset for further statistical analysis [156,157].

In this study, the relationship between the taxa richness and morphological diversity of each tested subfamily was not consistent. First, the genus/species richness of Pimeliinae and Stenochiinae (215/835 and 73/375) was higher than other subfamilies except Tenebrioninae (220/1198) (Table 1). Diaperinae showed the highest morphological diversity for pronotum (0.0295) and elytron (0.0066), and Pimeliinae (0.0267/0.0051) and Stenochiinae (0.0171/0.0042) also had relative high MD (Table 2). Among these subfamilies, Pimeliinae showed the highest tribe richness (33), followed by Tenebrioninae and Diaperinae (29/10), and Stenochiinae had the lowest (3). Second, the tribe/genus/species richness of Lagriinae (9/62/357) was higher than that of Alleculinae (2/54/317), but the morphological diversity of its pronotum was the opposite (0.0100/0.0111) (Figure 3). Phrenapatinae, Nilioninae, and Zolodininae had low tribe/genus/species richness (8/16 and 2/2), but the morphological diversity of the pronotum of Nilioninae (0.0130) and the elytron of Zolodininae and Phrenapatinae was not very low (0.0036/0.0026). Third, Tenebrioninae had high tribe/genus/species richness (29/220/1198), but the morphological diversity of the pronotum and elytron was within the average range for all subfamilies tested (0.0124/0.0041). Diaperinae had the highest species diversity index (0.0295/0.0066) but not the highest tribe/genus/species richness (13/43/286).

Diaperinae, Pimeliinae, and Stenochiinae were found to have high morphological diversity of pronotum (0.0295/0.0267/0.0171) and elytron (0.0066/0.0051/0.0042) (Table 2). However, the relationship between pronotum and elytron in the same taxonomic taxa was found to be not invariable: The subfamilies Tenebrioninae, Lagriinae, Zolodininae, and Phrenapatinae had lower MD of pronotum (0.0124/0.0100/0.0046/0.0074) than those in the elytron test (0.0041/0.0039/0.0036/0.0026); the morphological diversity of the pronotum of Nilioninae and Alleculinae (0.0130/0.0111) was higher than that of the elytron (0.0014/0.0019).

### 3.3. Correlation Analysis of Test Subfamilies

Spearman correlation analysis was conducted on the biodiversity index of the test taxa, and the relationship between the richness (each tribe/genus/species in the subfamily) and the morphological diversity of the test features was obtained (Table 3). First, the correlation between the morphological diversity of elytron and the richness of tribe/genus/species (0.782/0.600/0.571) was found to be higher than that in the pronotum test (0.521/0.350/0.500); second, there was a significant correlation between the morphological diversity of elytron and the richness of tribe/genus ($p = 0.006/0.044$). In addition, regression analysis of test parameters was carried out based on ordinary least square (OLS), and a scatterplot containing six best-fit lines was obtained (Figure 4A). The intercept value of test groups on the *X*-axis of the fitting line was more chaotic in the elytron test results, and the intercept

value of test groups on the *Y*-axis was more concentrated in the pronotum test results; the slope of the tribe-level test was higher than both the genus-level and species-level test. Furthermore, in the morphological space consisting of the diversity of elytron morphology and pronotum, Alleculinae, Diaperinae, Phrenapatinae, Pimelinae, Stenochiinae, and Tenebrioninae are concentrated on both sides of the fitting line (Figure 4B), and Zolodininae, Lagriinae, and Nilioninae subfamilies are dispersed around the fitting line.

## 4. Discussion

Based on the dataset of darkling beetles, we explored the correlation between the morphological diversity and taxa richness of the tribes/genera/species in each subfamily and found that the correlation between the richness of the genus/species and the morphological diversity of test features in the test of specific groups was positive; the correlation between test parameters in other subfamilies was found to be inconsistent. According to the results of the pronotum and elytron tests, we found that the correlation between test parameters decreased as the order decreased from tribe to species, and the correlation between MD and taxa richness was higher in the elytron test than the pronotum test.

Taxonomic richness and morphological diversity are often used to evaluate the degree of biological diversity, and their importance as a basis for further understanding the development of taxa has been confirmed [9,158]. In this study, we found that the richness of genus/species level Pimeliinae and Stenochiinae was positively correlated with the variation trend of morphological diversity of test features, which was consistent with previous findings [2]. However, correlations between test parameters were not consistent in other subfamilies of Tenebrionidae, such as Tenebrioninae, which is highly abundant, accounting for about one-third of the whole test dataset but show low morphological diversity in the test. This may be because of three aspects: First, according to the classical niche theory, because regional communities are mainly formed by interactions between species (e.g., the principle of competitive exclusion and limit similarity) [159], there will be higher competition in the same habitat, which will differentiate into more distinct levels or regional niches [160–165]. For example, most members of Diaperinae (e.g., Diaperini, Gnathidiini, and Hypophlaeini) live in forests with moderate humidity, except some tribes which live in arid places such as sandy areas or beaches (e.g., Crypticini, Hyociini, Phaleriini, Ectychini, and Trachyscelini) [11,123]. Second, the influence of interspecific competition pressure driving organisms produces different external forms for better use of the surrounding resources (such as food and habitat) in their process of evolution [166–168]. This competitive pressure causes organisms to change their existing form in order to meet similar functional requirements, and they become diversified [169,170]; for example, Diaperinae had the most varied forms but relatively low richness in this study. Third, the adaptive evolution of darkling beetles is closely related to their functional morphology, and the multiple origins of external that morphological features evolved in various environments make the test results more complex. Some members of Tenebrionidae live in desert and semi-desert areas, where the harsh environment leads to adaptive changes in structure and function, with many of the morphological variations resulting in certain functional preservation [43,171–174].

The correlation between morphological and functional diversity explains the differences among taxa to some extent in this study [175]. For instance, Tenebrioninae, Pimeliinae, and Lagriinae, typical species with healing of elytron, are rich in species diversity. In order to make better use of the surrounding environmental resources, darkling beetles living in different ecological environments evolved diverse eating habits, resulting in a more complex mechanism of biodiversity. In studies of muscular and skeletal versatility in shrews, researchers found that diverse foraging habits and morphologies produce similar functional outputs, leading to ecological convergence between populations and a link between diverse morphological features [169,170]. As typical omnivorous animals, tenebrionid beetles consume various foods, including fresh or withered plants, animal carcasses, feces, and humus. For example, the beetles of Blaptini and Opatrini are generally phytophagous,

feeding on the roots, leaves, flowers, and fruits of fresh or withered plants, and a few Blaptini insects prey on small insects [176]. This single foraging form makes this group very species abundant but with a low degree of morphological variation. However, the beetles of genus *Diaperis* under the Diaperini tribe live on the fruiting bodies of fungi, the arboreal *Ceropria* species mostly live under the bark of dead and rotting trees and feed on rotting trees infected by fungi, and warehouse-dwelling species often feed on damp, mildewed, and fungus-infected grains and have a wide diet [98]. Although the richness of these taxa in this study was not high, they had the highest morphological diversity.

Higher taxa are biological entities that are relatively easy to identify, and the origin of a new higher taxon has a long-term phylogenetic trend, involving the evolutionary changes of a large number of features [177,178]. The significance of higher taxa in the study of biodiversity patterns has been confirmed [179], and a strong correlation between the higher taxa and species composition of Coleoptera has also been revealed [180]. In this study, we also confirmed that the higher taxa have better applicability in biodiversity studies. Based on the morphological information of elytron, it was found that the correlation between taxa richness and morphological diversity was higher at the tribe level than at the genus or species level. However, this conclusion was not verified in the pronotum test. The correlation between the morphological diversity and genus richness of the pronotum was lower than the test results at the species level, while the correlation between tribe richness and morphological diversity was similar to that at the species level, which does not reflect the superiority of the higher taxa in biodiversity research. In addition, the study found that although the morphological diversity of elytron in all subfamilies was not as high compared to pronotum, the correlation between the morphological diversity of elytron and the richness of each taxon was higher. This may be due to the fact that the varied morphology of pronotum is affected by the darkling beetle's feeding behavior and living environment, and this speculation has also been confirmed in biological studies. Manseguiao found that different environmental conditions lead to different shapes of the pronotum of *Philippines reichei* between populations, which is important in defense against predators, parasite resistance, resource utilization, and competitiveness [24]. Taravati found that the gradient of environmental conditions leads to changes in the gradient of the pronotum of *Eliodontes* sp. [181]. Dela found that the nutrition of rice black bug resulted in a difference in the morphology of the pronotum [182]. The darkling beetles' pronotum provides mechanical protection, mimicry, and the exchange of visual signals with the same clan [24,183], and its internal muscles are also related to flight and movement [184,185], which enables it to adapt to diverse habitats. The variation of taxonomic morphology and feeding behavior under different habitats conditions is inconsistent, thus affecting the functional morphological development of the pronotum, so the diversity of ecological niches leads to the formation of the darkling beetles pronotum affected by more factors, which makes the variation more complex and unstable. The elytron can protect soft tissues on the notum of beetles and assist in flight and can heal and form sub-elytral cavities on the dorsal side of the abdomen in most Tenebrionidae [173,174]. With its stable morphology and simple function, elytron is more suitable for expressing the biodiversity of taxa.

## 5. Conclusions

Our study confirmed the applicability of higher categories in biodiversity research. We found positive correlations between the richness of the genus/species and the morphological diversity of the test features in the test of specific groups, and the correlation between test parameters decreased as the order decreased from tribe to species. However, during the experiment, we only controlled the coverage of the total sample, and only one image was collected for each species, which limited the objective to a certain extent. Therefore, more test features (such as foot and body color of darkling beetles) and a balanced sample size should be set to further explore the impact factors when objectively testing the relationships between biological indicators.

**Author Contributions:** Conceptualization, M.B. and X.W.; methodology, M.B. and Y.T.; sample collection, L.C. and Y.T.; sample identification and sorting, L.C. and Y.Z.; data analysis and processing, L.C. and Y.T.; writing—original draft, L.C.; writing—review and editing, Y.T., Z.S., F.M. and M.B. All authors have read and agreed to the published version of the manuscript.

**Funding:** This research was funded by the National Science & Technology Fundamental Resources Investigation Program of China (Grant Nos. 2019FY100400, 2019FY101800), the National Natural Science Foundation of China (No. 31961143002), the Bureau of International Cooperation, Chinese Academy of Sciences, GDAS Special Project of Science and Technology Development (Nos. 2020GDASYL-20200102021, 2020GDASYL-20200301003), the Second Tibetan Plateau Scientific Expedition and Research Program (STEP), Grant No. 2019QZKK05010101, and a project from Hainan Yazhou Bay Seed Lab (No. B21HJ0102).

**Institutional Review Board Statement:** Not applicable.

**Data Availability Statement:** The data presented in this study are available in article.

**Acknowledgments:** We are grateful to the Guodong Ren and Xingke Yang for the discussion and advices. We are also grateful to Yandong Chen for his photos of *Catapiestus* sp. (Tenebrionidae, Stenochiinae, Cnodalonini) specimens for Figure 1.

**Conflicts of Interest:** The authors declare no conflict of interest.

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
