# Peer review of "Study on the Relationship between Richness and Morphological Diversity of Higher Taxa in the Darkling Beetles (Coleoptera: Tenebrionidae)"

_diversity, doi:10.3390/d14010060_

Round 1

Reviewer 1 Report

This is a well written paper analysing a spectacular data set of one of the most diverse families of Coleoptera. I recommend its publication with only minor suggestions of adding the following information in the text:

1) In the table 1 it is recommended to include also percentage showing how complete was the data set within all tested taxonomic levels of all test groups. There is a general information in the first paragraph of the section 2.2 that 29% of the genera was sampled, but it would be good to know groups where the sampling is better and those when a more complete data set can be generate in future research.

2) It would be good to add a paragraph (in the Introduction) on advancement of studies on phylogeny of tenebrionid beetles, as this is crucial for understanding the results. If the phylogeny is not well established, it is difficult to build up any more elaborate hypotheses. 

And a single editing error - in the line 314 a space is lacking between words "elytral" and "cavity".

Author Response

  1. In the table 1 it is recommended to include also percentage showing how complete was the data set within all tested taxonomic levels of all test groups. There is a general information in the first paragraph of the section 2.2 that 29% of the genera was sampled, but it would be good to know groups where the sampling is better and those when a more complete data set can be generate in future research.

Author’s response: Thanks for the suggestions. We have added the corresponding data in Table 1 (see Line 130, please).

  1. It would be good to add a paragraph (in the Introduction) on advancement of studies on phylogeny of tenebrionid beetles, as this is crucial for understanding the results. If the phylogeny is not well established, it is difficult to build up any more elaborate hypotheses.

Author’s response: Thanks for the suggestions. We agree with this suggestion and have added the research progress on the phylogeny of tenebrionid beetles in the introduction. (see Line 80-92, please).

  1. And a single editing error - in the line 314 a space is lacking between words "elytral" and "cavity".

Author’s response: We agree with the reviewer. We have added a space between the two words (see Line 353, please).

Reviewer 2 Report

The article is an interesting study from the point of view of evolutionary biology. The research results are interesting for working out the methodology for assessing ecosystems by model groups of insects.

At the same time, the article contains a number of shortcomings that need to be eliminated:

  • for some reason, the authors limited themselves to the species level of diversity of darkling beetles, without analyzing the literature on the morphological diversity of darkling beetle species at the population level, but these data would be very interesting for this study;
  • in Table 1 it is necessary to add three columns containing information about the% of species, genera and tribes considered in this article in relation to their world diversity;
  • in figures (for example, in Figure 3) it is better not to use bold type;
  • it is better to round off all numbers to thousandths on the right ordinate in Figure 3;
  • it is incorrect to “build on” the characteristics one above the other in Figure 3; they need to be depicted next to each other;
  • I consider the calculation of the diversity index for the subfamily Zolodininae (for 2 species) incorrect (Table 2);
  • subfamilies everywhere (Table 2, Figures 2, 3 and others) must be arranged according to international taxonomy (http://doi.org/10.3897/zookeys.1050.64217), and not alphabetically;
  • it is necessary to indicate the reliability for each of the correlation coefficients in Table 3;
  • it is not necessary to write all words with a capital letter (for example, in Table 3 or in Figure 3);
  • all fonts in all figures should be approximately equal in size to the fonts of the main text of the article; on some of the figures, for example, in Figure 1, they are gigantic, on another part, for example, in Figure 2, they are very small, practically unreadable;
  • I believe that Figure 2 is the main result of research: it needs to be made at least twice as large;
  • all % in the article should be rounded to tenths, and not as the authors like (to whole – for example, line 108, 148 or to thousandths – for example, line 145, 146, 161, abscissa and ordinate axes of Figure 2);
  • the trend lines in Figure 4 must be removed, they do not reflect the location of the points; in general, Figure 4 is very small, unreadable; all numbers must be rounded to hundredths on the ordinate axis of Figure 4; it is possible to make a logarithmic scale on the abscissa of Figure 4a;
  • it is better to indicate a specific value of P in the text of the article, and not be limited to information that it is less than 0.05 (for example, line 217);
  • Are the values of the characteristics discussed in the Results of the article random (for example, in Figure 3)? I recommend using the Chi-square Test or the Lambda Test to assess differences from the null hypothesis; it is the central element of the article, which has not been analyzed by correct statistical methods;
  • for some reason, figures and tables are placed not in the Results, where they should be, but in Materials and methods;
  • References were not edited according to the rules of the journal, with very many errors (on average, three to four errors per line): in the titles of articles it is not necessary to write all words with a capital letter (for example, lines 341, 343, 350, 382), the journal titles must be abbreviated according to the standard (for example, lines 365, 393, 394, 641, 642), there are no journal titles (for example, lines 360, 367, 402), incorrect punctuation (for example, lines 404, 407), Latin names are not italicized (for example, lines 400, 409), for some reason, the month is indicated (for example, line 421, 427, 437, 443), errors in the journal titles (for example, line 447), pages are missing (for example, line 449, 562), the editor, publisher and city are not indicated (for example, line 469, 471 ), pages are incorrectly specified (for example, lines 474, 504) and many other types of errors;
  • the bibliography does not contain DOI references; they must be added;
  • Sources 128–145 are best removed from the bibliography by moving references to sites in the text of the article.

The article needs significant revision; however, the results are interesting and can be recommended for publication.

Author Response

  1. for some reason, the authors limited themselves to the species level of diversity of darkling beetles, without analyzing the literature on the morphological diversity of darkling beetle species at the population level, but these data would be very interesting for this study.

Author’s response: Thanks for the suggestions. Biodiversity indicators of geographic populations in local areas are affected by many factors (such as habitat and niche changes), which make the study of populations more complicated. Furthermore, the relationship between biodiversity parameters also fluctuates depending on the changes in test populations and test conditions. Therefore, this study is based on a global dataset to explore the relationship between the richness and morphology of higher taxa, we attempt to explore the trends between biodiversity indicators from another perspective, rather than focusing on diversity at the population level. In addition, in this paper, we not only discuss the relationship between taxonomic richness and morphological diversity, but also compare the morphological changes of test features (pronotum and elytron), aiming to reveal differences in biological and morphological evolution of different taxa.

  1. in Table 1 it is necessary to add three columns containing information about the% of species, genera and tribes considered in this article in relation to their world diversity.

Author’s response: Thanks for the suggestions. We have added the number of tribes and genera described in Table 1 (see Line 130, please). The large distribution and large number in nature, the complex living habits and the diverse external forms make it difficult for current research of darkling beetles to be systematically and accurately sorted out at the higher-level (subfamily, tribe, genus). Most researchers focus on the study of multiple populations in local areas, due to the inconsistency of morphology and developmental concepts, the systematic status of many genera has changed frequently. Therefore, we need to face a situation that it is difficult to sort out a widely accepted set of data count: including accurate data for each category until now, which is especially prominent in the species-level statistics. Therefore, in the previous manuscript we did not list the exact number of the world described species. In addition, this paper will focus on the trend of correlation between the biodiversity indices of the higher category, so in order to show the degree of morphological diversity of different taxa more objectively, we try to obtain a higher proportion of the number of test groups in principle of sampling, most categories get a high sampling rate (see Line 130, please). At the same time, in this manuscript, we have also added the objectivity verification of morphological data to ensure that the results obtained are more objective (see Line 215 to Line 223, please).

  1. in figures (for example, in Figure 3) it is better not to use bold type;

it is better to round off all numbers to thousandths on the right ordinate in Figure 3.

Author’s response: We agree with the reviewer. We have changed the bold text in all the graphs to the regular type and round off all numbers on the right ordinate in Figure 3 to thousandths.

  1. it is incorrect to “build on” the characteristics one above the other in Figure 3; they need to be depicted next to each other.

Author’s response: Thanks for the great comments. After discussion, we found that the author's misunderstanding led to the incorrect composition of Figure 3. It has been modified now. Thanks again for pointing out this mistake (see Line180, please).

  1. I consider the calculation of the diversity index for the subfamily Zolodininae (for 2 species) incorrect (Table 2).

Author’s response: Thanks for the suggestions. We agree that the diversity index of the subfamily Zolodininae is not 2, the 2 in the table is just the sampling number in this study. This subfamily is very small (only 3 genera be found over the world), very few published individual records make us have to do our best to collect morphological information about these specimens. In the sample information that has been published so far, we have selected 2 specimens (which meet the geometric morphological analysis methods for this study), which is used to represent the morphological information of this subfamily, so as to make the biodiversity study of Tenebrionidae more comprehensive. In subsequent studies, we will continue to add relevant kinds of information to obtain more objective results. Thanks again for your suggestions.

  1. subfamilies everywhere (Table 2, Figures 2, 3 and others) must be arranged according to international taxonomy (http://doi.org/10.3897/zookeys.1050.64217), and not alphabetically.

Author’s response: Thanks for the suggestions. We have revised the order of all subfamilies in accordance with the international taxonomy.

  1. it is necessary to indicate the reliability for each of the correlation coefficients in Table 3.

Author’s response: We agree with the reviewer. We have increased the reliability of the correlation coefficients in Table 3 (see Line 167-168, please).

  1. it is not necessary to write all words with a capital letter (for example, in Table 3 or in Figure 3).

Author’s response: We agree with the reviewer. We have normalized the words in Table 3 and Figure 3 (see Line 167 and Line 180, please).

  1. all fonts in all figures should be approximately equal in size to the fonts of the main text of the article; on some of the figures, for example, in Figure 1, they are gigantic, on another part, for example, in Figure 2, they are very small, practically unreadable.

Author’s response: Thanks for the great comments. We have modified all the figures to make sure that the fonts on the current images are legible.

  1. I believe that Figure 2 is the main result of research: it needs to be made at least twice as large.

Author’s response: We agree with the reviewer. We have adjusted the size of figure 2 to make it readable (see Line 174, please).

  1. all % in the article should be rounded to tenths, and not as the authors like (to whole – for example, line 108, 148 or to thousandths – for example, line 145, 146, 161, abscissa and ordinate axes of Figure 2).

Author’s response: Thanks for the suggestions. We felt the % can more intuitively reflect the parameter differences between taxa, and this percentage display form is also a common format in geometric morphology research.

  1. the trend lines in Figure 4 must be removed, they do not reflect the location of the points; in general, Figure 4 is very small, unreadable; all numbers must be rounded to hundredths on the ordinate axis of Figure 4; it is possible to make a logarithmic scale on the abscissa of Figure 4a.

Author’s response: Thanks for the suggestions. We agree with the reviewer, and have removed the trend line in Figure 4A and resized Figure 4 to make it readable. About the last comment (as for the idea of making a logarithmic scale on the horizontal coordinate), the scale of the current horizontal axis reflects the true value of each taxa richness, and we believe the readers may get the degree of correlation between the two test parameters more intuitively by the current display format.

  1. it is better to indicate a specific value of P in the text of the article, and not be limited to information that it is less than 0.05 (for example, line 217).

Author’s response: We agree with the reviewer, and we have also added the specific value of P in the corresponding position in the text (see the Line 256, please).

  1. Are the values of the characteristics discussed in the Results of the article random (for example, in Figure 3)? I recommend using the Chi-square Test or the Lambda Test to assess differences from the null hypothesis; it is the central element of the article, which has not been analyzed by correct statistical methods.

Author’s response: Thanks for the suggestions. Since the morphological diversity index involved in this study is obtained by quantifying the test features, the diversity of the relevant parameters reflects the distribution of semi-landmarks on the features of each sample in space and the difference between other homologous feature semi-landmarks, the differences in this quantitative morphological space are difficult to set the expected value, so it cannot be well verified by means such as Chi-Square test. In this new version of our manuscript, we analyze the test points to test whether the variation in shape among a set of specimens is too large and thus statistical methods based on the tangent space approximation cannot be used based on the tps-small software, by collating the test points of the features (see Line 168-170, please). We found the correlation (uncentred) between the tangent space, Y, regressed onto Procrustes distance (geodesic distances in radians) was 0.999935/0.999998 for pronotum test and elyron group, which indicate little doubt on the basis of the results and proved the acceptability of the data set for further statistical analysis (see Line 214-223, please).

  1. for some reason, figures and tables are placed not in the Results, where they should be, but in Materials and methods.

Author’s response: Thanks for the suggestions. Because of all the figures and tables we first mentioned in this article were in the part of materials and methods, we included them in there for ease of reading.

  1. References were not edited according to the rules of the journal, with very many errors (on average, three to four errors per line): in the titles of articles it is not necessary to write all words with a capital letter (for example, lines 341, 343, 350, 382), the journal titles must be abbreviated according to the standard (for example, lines 365, 393, 394, 641, 642), there are no journal titles (for example, lines 360, 367, 402), incorrect punctuation (for example, lines 404, 407), Latin names are not italicized (for example, lines 400, 409), for some reason, the month is indicated (for example, line 421, 427, 437, 443), errors in the journal titles (for example, line 447), pages are missing (for example, line 449, 562), the editor, publisher and city are not indicated (for example, line 469, 471 ), pages are incorrectly specified (for example, lines 474, 504) and many other types of errors.

Author’s response: We agree with the reviewer. We have checked and revised all references as required.

  1. the bibliography does not contain DOI references; they must be added;

Author’s response: Thanks for the suggestions. We have added the DOI references to the bibliography (for some Chinese Journal, it doesn’t have a DOI) as required.

  1. Sources 128–145 are best removed from the bibliography by moving references to sites in the text of the article.

Author’s response: We have revised the references in the corresponding parts as suggested.
